# AUTOMATIC CURRICULUM GENERATION FOR REINFORCEMENT LEARNING IN ZERO-SUM GAMES

## ABSTRACT

Curriculum learning (CL), whose core idea is to train from easy to hard, is a popular technique to accelerate reinforcement learning (RL) training. It has also been a trend to automate the curriculum generation process. Automatic CL works primarily focus on goal-conditioned RL problems, where an explicit indicator of training progress, e.g., reward or success rate, can be used to prioritize the training tasks. However, such a requirement is no longer valid in zero-sum games: there are no goals for the agents, and the accumulative reward of the learning policy can constantly fluctuate throughout training. In this work, we present the first theoretical framework of automatic curriculum learning in the setting of zero-sum games and derive a surprisingly simple indicator of training progress, i.e., the Q-value variance, which can be directly approximated by computing the variance of value network ensembles. With such a progression metric, we further adopt a particle-based task sampler to generate initial environment configurations for training, which is particularly lightweight, computation-efficient, and naturally multi-modal. Combining these techniques with multi-agent PPO training, we obtain our final algorithm, *Zero-sum Automatic Curriculum Learning* (ZACL). We first evaluate ZACL in a 2D particle-world environment, where ZACL produces much stronger policies than popular RL methods for zero-sum games using the same amount of samples. Then we show in the challenging hide-and-seek environment that ZACL can lead to all four emergent phases using a single desktop computer, which is reported for the first time in the literature. The project website is at https://sites.google.com/view/zacl.

## 1 INTRODUCTION

Curriculum learning (CL) (Bengio et al., 2009), whose core idea is to generate training samples from easy to hard, is a popular paradigm to accelerate the training of reinforcement learning (RL) agents (Lazaric et al., 2008; Taylor et al., 2008; Narvekar et al., 2020). Starting from simple tasks, an RL agent can progressively adapt to tasks with increasing difficulty according to a properly designed curriculum, and finally solve hard tasks with fewer samples than naive RL training from scratch with uniformly sampled training tasks.

Automating the design of curriculum for RL training has attracted much research interest. An ordinary formulation is the teacher-student framework (Matiisen et al., 2019; Portelas et al., 2020), where a "teacher" proposes task configurations that are neither too easy nor too hard for the "student" RL agent to solve. A key ingredient in order to generate suitable task configurations is to measure the progress of the learning student. In goal-oriented RL problems or cooperative multi-agent games, the training progress is straightforward to measure since the success rate for reaching a goal or the accumulated reward can explicitly reflect the current performance of the student on a specific task (Wang et al., 2019; Florensa et al., 2018; Chen et al., 2021).

However, an explicit progression metric does not exist in the setting of zero-sum games, where the ultimate goal of the RL agent is no longer reaching any goal or getting high rewards. Instead, the convergence criterion is to find a Nash equilibrium (NE) efficiently. The accumulated reward of a policy for one player would oscillate when it is exploiting or being exploited by its opponent throughout RL training. Therefore, it becomes non-trivial to measure how "close" the current policy to an NE is. Existing curricula in zero-sum games are typically based on heuristics, i.e., training on

an increasing number of agents (Long et al., 2020; Wang et al., 2020b) or adapting other environment parameters according to domain knowledge (Berner et al., 2019; Tang et al., 2021). It remains unclear how to automate the generation process of these task parameters.

In this work, we propose a novel automatic CL framework for multi-agent RL training in zero-sum games. We theoretically derive a surprisingly simple progress metric, i.e., Q-value variance, as an implicit signal of the learning progress towards an NE. By prioritizing learning on game configurations with high Q-value variance, we are implicitly tightening a lower bound of the true distance between the learning policy and a desired NE. This simple metric is also straightforward to incorporate into a configuration sampler to automatically generate the training curriculum for the RL agents to accelerate convergence towards NE.

We approximate the Q-value variance by value variance and implement it as the empirical uncertainty over an ensemble of independently learned value networks. We then develop a curriculum generator that samples game configurations according to an empirical probability distribution with density defined by the value uncertainty. In order to keep track of the constantly evolving and multi-modal density landscape induced by value uncertainty throughout RL training, we adopt a non-parametric sampler that directly samples configurations from a diversified data buffer. Combining the progression metric and non-parametric curriculum generator with a multi-agent RL backbone MAPPO (Yu et al., 2022), we derive our overall automatic curriculum learning algorithm for zero-sum games, *Zero-sum Automatic Curriculum Learning* (ZACL). We first evaluate ZACL in a 2D particle-world benchmark, where ZACL learns stronger policies with lower exploitability than existing multi-agent RL algorithms for zero-sum games given the same amount of environment interactions. We then stress-test the efficiency of ZACL in the challenging hide-and-seek environment. ZACL produces the emergence of all four phases of strategies only using a single desktop machine, which is reported for the first time. Moreover, ZACL also consumes substantially fewer environment samples than large-scale distributed PPO training.

## 2 RELATED WORK

Curriculum learning has a long history of accelerating RL training (Asada et al., 1996; Soni & Singh, 2006; Lazaric et al., 2008; Taylor et al., 2008; Narvekar et al., 2020). In the recent literature, automatic curriculum learning (ACL) is often applied to the goal-oriented RL setting where the RL agent needs to reach a specific goal in each episode. ACL methods design or learn a smart sampler to generate proper task configurations or goals that are most suitable for training advances w.r.t. some progression metric (Florensa et al., 2017; 2018; Racaniere et al., 2019; Matiisen et al., 2019; Portelas et al., 2020; Dendorfer et al., 2020; Chen et al., 2021). Such a metric typically relies on an explicit signal, such as the goal-reaching reward or reward changes (Wang et al., 2019; Florensa et al., 2018; Portelas et al., 2020; Matiisen et al., 2019), success rates (Chen et al., 2021), and the expected value on the testing tasks. However, in the setting of zero-sum games, these explicit progression metrics become no longer valid since the reward associated with a Nash equilibrium can be arbitrary.

There are also ACL works utilizing reward-agnostic metrics. One representative category of methods is asymmetric self-play (Sukhbaatar et al., 2018; Liu et al., 2019; OpenAI et al., 2021), where two separate RL agents are trained via self-play with one agent setting up goals to exploit the other. Such a self-play training framework implicitly constructs a competitive game leading to an emergent curriculum for the training agents. Some recent works have also shown that this training process can be provably related to regret minimization with proper changes (Dennis et al., 2020; Gur et al., 2021). However, these methods still assume a goal-oriented problem and it remains non-trivial to adapt them to zero-sum games. (Zhang et al., 2020) adopts a similar value disagreement ACL metric to tackle a collection of single-agent goal-reaching tasks, which is perhaps the most technically similar work to our method. In addition to different problem domains of interest, we remark that (Zhang et al., 2020) develops the method in a purely empirical fashion while we follow a complete theoretical derivation. It is a beautiful coincidence that empirical observations and theoretical analysis converge to the same implementation even with parallel motivations.

Applying RL to solve zero-sum games can be traced back to the TD-Gammon project (Tesauro et al., 1995) and has led to great achievements in defeating professional humans in complex competitive games (Jaderberg et al., 2018; Berner et al., 2019; Vinyals et al., 2019). Most RL methods for zero-sum games can be proved to converge to an (approximate) Nash equilibrium (NE) or correlated

equilibrium. A large body of works is based on regret minimization (Brown et al., 2019; Steinberger et al., 2020; Gruslys et al., 2020), which assumes game rules are known. Another notable line of works adopts model-free RL algorithms with self-play training, such as fictitious self-play (FSP) (Heinrich et al., 2015; Heinrich & Silver, 2016) and its variants (Jin et al., 2021; Bai et al., 2020; Perolat et al., 2022), which derives a single NE policy. It is often empirically observed that the RL agent will obtain increasingly more complex skills during the self-play training, which is often called the emergent auto-curricula (Bansal et al., 2018; Baker et al., 2020). Our work focuses on how to accelerate self-play training to reach an NE faster rather than analyzing emergent behaviors. The self-play framework can be further extended to population-based training (PBT) by maintaining a policy pool and repeatedly adding best-response policies to it, which is called the double oracle method (McMahan et al., 2003) or policy space response oracles (PSRO) (Lanctot et al., 2017). There are also PSRO-specific ACL methods for faster best-response training by constructing a smart mixing strategy over the policy pool according to the policy landscape (Balduzzi et al., 2019; Perez-Nieves et al., 2021; Liu et al., 2021). By contrast, our method focuses on curriculum generation over *low-level states* or *task configurations* and is complementary to these meta-level CL methods. Besides, some CL methods for zero-sum games adopt domain heuristics over the number of agents (Long et al., 2020; Wang et al., 2020b) or environment specifications (Berner et al., 2019; Tang et al., 2021) while our method only assumes a minimal requirement of a generic state sampler.

Our work adopts a non-parametric state sampler which is fast to learn and naturally multi-modal, instead of training an expensive deep generative model like GAN (Florensa et al., 2018). Such an idea has been recently popularized in ACL literature. Representative samplers include Gaussian mixture model (Warde-Farley et al., 2019), stein variational inference (Chen et al., 2021), Gaussian process (Mehta et al., 2020), or simply evolutionary computation (Wang et al., 2019; 2020a). Technically, our method is also related to prioritized experience replay (Schaul et al., 2015; Florensa et al., 2017; Li et al., 2022) with the difference that we maintain a diversified buffer (Warde-Farley et al., 2019) to approximate the uniform distribution over the state space.

## 3 CURRICULUM LEARNING FOR ZERO-SUM GAMES

We consider decentralized Markov decision process (Dec-MDP) denoted as $(N, \mathcal{S}, \mathcal{A}, R, P, \rho_0, \gamma)$, where $N$ is the number of agents, $\mathcal{S}$ and $\mathcal{A}$ are state and action space shared across all the agents, $R_i, i \in [1, N]$ are reward functions for different agents, $P$ denotes the transition probability, $\rho_0$ is the distribution of initial states, and $\gamma$ is the discount factor. Each agent optimizes its policy $\pi_i$ with multi-agent reinforcement learning. We focus on mixed-cooperative competitive zero-sum games in particular, where $n$ out of the total $N$ agents share the same reward function $R$, and the remaining $N-n$ agents are with the reward function $-R$. For simplicity, we denote the joint policy distribution for the team of agents with reward $R$ as $\pi_+$ and the joint policy for other agents as $\pi_-$. In this way, the problem setting becomes a standard two-player zero-sum game. The joint policy over the two teams of agents is denoted as $\boldsymbol{\pi}$, and the policy of either of the two teams is denoted as $\pi_{\pm}$.

Curriculum learning assumes a task space $\mathcal{T}$ the agent is allowed to be trained over, which can be collections of initial states, environment parameters, or goals. In automatic curriculum learning, a task generator typically draws samples from $\mathcal{T}$ for the RL agent to train on according to the current performance of the policy. Without loss of generality, we assume the state $s$ contains all necessary information to construct a task configuration. Therefore, a good curriculum is equivalent to a smart state sampler proposing appropriate states to initialize each episode. So, in the following content, we use the terminology of *state* to denote the concept of *task configurations* in curriculum learning.

### 3.1 MULTI-AGENT RL AS DISTRIBUTION MATCHING

In non-cooperative Markov games, it is difficult to directly evaluate the learning progress with respect to the reward function obtained by each individual agent. In principle, though, the learning progress could be modeled via some statistical distance between the trajectory distribution of the current agents, and that of the agents under some optimality notion, similar to a multi-agent imitation learning setting (Song et al., 2018; Yu et al., 2019). However, even this is hard to characterize, as we do not have access to the trajectories of the optimal agents. Ideally, we should devise our curriculum from the policies and/or value functions alone.

Our first step is to connect this statistical distance with value functions. Under reasonable assumptions such as bounded rationality (McKelvey & Palfrey, 1995; 1998), it is possible to characterize the joint distribution of trajectories generated by agents at some equilibrium, given the Q-functions. For example, we can consider the Logistic Stochastic Best Response Equilibrium (LSBRE) introduced in Yu et al. (2019).

**Definition 1** (LSBRE). *The logistic stochastic best response equilibrium (LSBRE) for a Markov game with horizon $T$ is a sequence of stochastic policies defined recursively via a Markov chain. The state of the Markov chain at step $k$ for each agent at time $t$ is $\{z_i^{t,(k)} : \mathcal{S} \to \mathcal{A}_i\}_{i=1}^N$, with each random variable $z_i^{t,(k)}(s)$ taking values in $\mathcal{A}_i$. The transition from step $k$ to $(k+1)$ is defined as:*

$$z_i^{t,(k+1)}(s^t) \sim P_i^t(a_i^t | \boldsymbol{a}_{-i}^t = \boldsymbol{z}_{-i}^{t,(k)}(s^t), s^t) = \frac{\exp\left(\lambda Q_i^{\boldsymbol{\pi}^{t+1:T}}(s^t, a_i^t, \boldsymbol{z}_{-i}^{t,(k)}(s^t))\right)}{\sum_{a_i'} \exp\left(\lambda Q_i^{\boldsymbol{\pi}^{t+1:T}}(s^t, a_i', \boldsymbol{z}_{-i}^{t,(k)}(s^t))\right)}, \quad (1)$$

*recursively from $t = T$ to $t = 0$, and $\lambda \in \mathbb{R}^+$ controls the level of rationality of the agents. Each joint policy $\boldsymbol{\pi}$ at time $t$ is given by $\boldsymbol{\pi}^t(a_1, \cdots, a_N | s^t) = P\left(\bigcap_i \{z_i^{t,(\infty)}(s^t) = a_i\}\right)$ where the probability is taken with respect to the unique stationary distribution of the $t$-th Markov chain.*

Informally, the LSBRE is the stationary distribution of a Markov chain over the joint policy $\boldsymbol{\pi}$, whose transition kernel at step $(k+1)$ given all agent policies at step $t$ is defined for every agent $\pi_i$ as the optimal maximum entropy RL policy if all other agents are acting according to their policies at step $k$. This equilibrium notion naturally specifies how agents would act under a certain reward function (with maximum entropy regularization controlled by $\lambda$), which also induces a joint distribution over the trajectory of all agents. In principle, we can then measure the learning progress in non-cooperate Markov games via some statistical distance between our current joint distribution versus the joint distribution induced by LSBRE. A common choice is the KL divergence: $\min_{\boldsymbol{\pi}} D_{\mathrm{KL}}(p(s,a) || \tilde{p}(s,a))$, where $p(s,a) = \boldsymbol{\pi}(a|s) \sum_{t=1}^T \Pr(s_t = s | \boldsymbol{\pi})$ is the occupancy measure induced by our parametrized policies $\boldsymbol{\pi}$, and $\tilde{p}(s,a)$ is the one induced by the LSBRE-optimal policies. Naturally, our policies reach the LSBRE when this objective is zero.

However, for computational efficiency, we often would like to take a pseudolikelihood approach (Besag, 1975), where we approximate the joint likelihood $\boldsymbol{\pi}(a|s)$ with a product of the conditionals, *i.e.*, $\pi_+^t(a_+|\boldsymbol{a}_-, s)\pi_-^t(a_-|\boldsymbol{a}_+, s)$. Intuitively, we replace the KL divergence of the joint policy with KL divergences of individual policies, and thus we replace the optimization over the joint policy with the optimization over the individual policies, where each agent updates its own policy according to the current fixed policy of other agents (as opposed to optimizing jointly). According to the LSBRE setup, each team operates under a maximum entropy (MaxEnt) RL-like setup (Haarnoja et al., 2017) conditioned on the behavior of other agents, and thus we can treat them as part of the environment, such that:

$$\pi_{\pm}^{\star}(a_{\pm}|s) = \frac{\exp\left(\lambda Q_{\pm}^{\star}(s,a)\right)}{\sum_{a_{\pm}'} \exp\left(\lambda Q_{\pm}^{\star}\left(s, a_{\pm}'\right)\right)}, \quad (2)$$

where $Q_{\pm}^*$ is the soft $Q$-function for either team. which aims to maximize the expected cumulative discounted reward along with the entropy in each state. Thus, we may represent a policy with a corresponding energy-based formulation $Q_{\pm}$, i.e., $\pi_{\pm}(a_{\pm}|s) = \exp\left(Q_{\pm}(s, a_{\pm})\right)$, such that $\pi_{\pm}$ is the optimal entropy-regularized policy under the normalized value function $Q_{\pm}$. As the the policy or the value function are tied in MaxEnt RL, means that we may represent the learning progress with either one of the quantities.

## 3.2 CURRICULUM FROM Q-VALUE VARIANCE

In curriculum learning, we wish to select experiences (state-action pairs) that would maximize the learning progress of the agents. Although we can theoretically measure the learning progress with KL divergence to a LSRBE-optimal policy, we have no access to that optimal policy in practice, so we have to rely on alternative metrics to decide the curriculum learning strategy.

For each agent, let us consider the case of learning $m$ value functions (denoted as $Q_{\pm}^{(1)}, Q_{\pm}^{(2)}, \ldots, Q_{\pm}^{(m)}$) in parallel; each of which induces a policy $\pi_{\pm}^{(j)}(a|s) = \exp(Q_{\pm}^{(j)}(s,a))$ (for

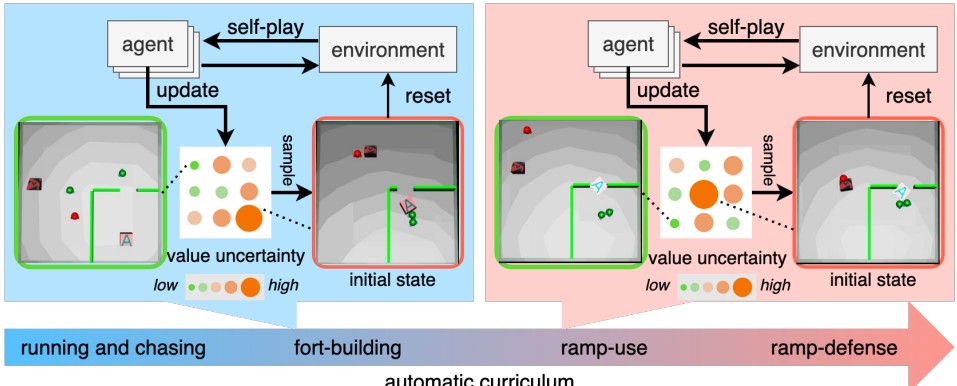

Figure 1: Illustration of ZACL in the hide-and-seek environment. In *Fort-Building* stage, the states with hiders near the box are of high uncertainty (red) and the agent can easily learn to build a fort by practising on these states, while the states with randomly spawned hiders are of low uncertainty (green) and contribute less to learning. By sampling the environment initial state from the density defined by the value uncertainty, the agent can proceed to new stages efficiently.

$j \in [1, m]$), and with infinite training capacity, they should all converge to the optimal policy (KL divergence becomes zero). Effectively, each agent is maximizing the following[1]:

$$\mathbb{E}_{\pi_\pm} \left[ Q_\pm^\star(s, a_\pm) - \log \pi_\pm(a|s) \right] \tag{3}$$

where $Q^\star$ is the soft $Q_\pm$-function that induces the optimal policy (Ziebart et al., 2008) for either team of agents. In curriculum learning, we need to prioritize over more useful state-action pairs to achieve the fastest learning progress. However, as we do not know the optimal policy distribution, we use a surrogate to decide where we could achieve better gains, and thus use it to decide our curriculum learning strategy. Intuitively, our strategy is three fold. First, we use multiple independent policies, from which we can obtain an average policy; we will prove that the average policy should perform better than individual policies acting independently. Second, for each state-action pair we characterize this performance gap, which we will tie with the disagreement of the individual policies/values. Third, if the disagreement is large, then the gap between the individual policies and the average policy is also large, and the gap with the (unknown) optimal policy should also be large. Therefore, we can use the disagreement of the individual policy values to devise a curriculum.

Define $\bar{\pi}(a|s) = \frac{1}{m} \sum_{j=1}^m \pi^{(j)}(a|s)$ as the average policy produced by the $m$ individual ones, then for any state-action pair $(s, a_\pm)$ we have the following:

**Proposition 1.** *For all policies $\pi_\pm^{(j)}$ and $Q_\pm^{(j)}$ such that $\pi_\pm^{(j)} = \exp(Q_\pm^{(j)})$:*

$$\frac{1}{m} \sum_{j=1}^m \pi^{(j)}(a_\pm|s)[Q_\pm^\star(s, a_\pm) - \log \pi^{(j)}(a_\pm|s)] \leq \bar{\pi}(a_\pm|s)[Q_\pm^\star(s, a_\pm) - \frac{1}{m} \sum_{j=1}^m Q_\pm^{(j)}(s, a_\pm)].$$

We list the proof in Appendix A.2, which uses two Jensen's inequalities. The differences between the two ends of the Jensen inequality is known as a Jensen gap:

$$\frac{1}{m} \sum_{j=1}^m \exp(Q_\pm^{(j)}(s, a_\pm))Q_\pm^{(j)}(s, a_\pm) - \frac{1}{m} \sum_{j=1}^m \exp(Q_\pm^{(j)}(s, a_\pm))\frac{1}{m} \sum_{j=1}^m Q_\pm^{(j)}(s, a_\pm), \tag{4}$$

which is essentially the covariance between $\exp(Q_\pm^{(j)}(s, a_\pm))$ and $Q_\pm$ (under the empirical distribution)[2]. The Jensen gap becomes zero if all the $Q$-values are equal, and maybe larger if the $Q$-values are more different. If we initialize the different functions randomly, then it is unlikely that they will become identical without having reached the optimal one.

---

[1]For conciseness, we use $\lambda = 1$, which corresponds to a certain degree of entropy regularization, but the argument is still valid for other $\lambda > 0$.

[2]As covariance is non-negative, this provides another proof angle for the proposition.

Our curriculum strategy is to choose state-action pairs that maximize the Jensen gap (which indirectly maximizes the gap between the individual policies and the optimal one), so intuitively we should select cases where the value functions have a high disagreement with each other. To further simplify the computation, we can use Taylor approximation to connect the Jensen gap to the variance of $Q_{\pm}$ (see Appendix A.1 for details).

## 4 METHOD

In this section, we describe the algorithmic components of ZACL. We first introduce value variance as a progress measurement in Sec. 4.1. Then we propose a non-parametric task sampler that samples the most feasible tasks according to the training progress in Sec. 4.2. In Sec. 4.3, we summarize the framework and the implementation details. An overview of ZACL is illustrated in Fig. 1.

### 4.1 VARIANCE ESTIMATION

Following the theoretical analysis, the uncertainty of value functions can be an approximate metric of learning progress. We train an ensemble of value functions $\boldsymbol{V} = \{V_{\psi_i}\}_{i=1}^m$ and take the statistical variance of their predictions on each state as the uncertainty measurement, $w(s) = \mathrm{Var}(\boldsymbol{V}(s))$. The value networks in the ensemble are initialized with different random parameters. In RL training, they are trained over mini-batches using different random splits from the same environment samples.

A naive way to compute the mean policy is to train separate policies for each value function over disjoint data batches that are used to train the value ensemble. However, maintaining $m$ policies is computationally expensive. Considering that we only care about learning one good policy instead of an ensemble, we instead train a single policy network with respect to the mean value function in practice. Taking the average of the value ensemble leads to a more accurate value estimation than each individual value function and therefore can accelerate policy learning.

### 4.2 STATE GENERATION FROM A DIVERSIFIED BUFFER

With value uncertainty as an approximate progression metric at hand, we can then treat it as the density distribution from which to sample promising states for the RL agent to train on. Training a neural generative model $f_\theta$ from which to sample states for automatic curriculum learning is a typical solution (Dendorfer et al., 2020; Fang et al., 2021), but it requires a large number of samples to fit accurately, therefore cannot adapt instantly to the ever-changing density (value uncertainty) in our case. Moreover, the value uncertainty distribution is highly multi-modal, which is hard to capture for many neural generative models.

We instead adopt a particle-based approach to efficiently sample from the value uncertainty distribution. We maintain a large diversified state buffer $\mathcal{Q}$ using all the visited states throughout RL training to approximate the state space, then draw samples from it with probability proportional to the value uncertainty associated with each state. Since the number of states we can store is limited while the state configuration space is exponentially large, it is important to only keep representative states so as to ensure good coverage of the state space with limited samples. When inserting new visited states into the buffer, we only keep those states sufficiently far from existing states.

### 4.3 ZERO-SUM AUTOMATIC CURRICULUM LEARNING

Combining the uncertainty-based curriculum progression metric and the efficient state generator with an RL backbone, we get our final algorithm ZACL for accelerating RL training in zero-sum games. When each episode resets, we use the state generator to sample good initial game configurations. After collecting a bunch of environment samples, we perform policy and value training and update the density estimation for all the existing samples in the state buffer with new value uncertainty. The novel states from rollout are then stored into the diversified state buffer.

**Implementations**: We adopt a state-of-the-art multi-agent RL algorithm MAPPO (Yu et al., 2022) as the backbone. Although our theoretical analysis follows the max-entropy RL framework, ZACL can empirically work with standard multi-agent RL algorithms. To avoid lack of exploration, we allow each episode to start from random states with 30% probability and generated states from the

| **Algorithm 1:** The ZACL Algorithm | **Algorithm 2:** State buffer update rule |
|---|---|

**Algorithm 1: The ZACL Algorithm**

**Input:** Empty state buffers $\mathcal{Q}$ and $\mathcal{M}$,
 probability to reset from state buffer $P_{exp}$,
 policy $\pi_\theta$, value ensemble $\{V_{\psi_i}\}_{i=1}^m$;
**Output:** final policy $\pi_\theta$;
**repeat**
    $s_0 \sim \mathcal{Q}$ if `rand()` $< P_{exp}$ else $s_0 \sim \rho_0$;
    Rollout $\pi_\theta$;
    Train $\pi_\theta, \{V_{\psi_i}\}_{i=1}^m$ via MARL;
    // Compute uncertainty
    $w_t \leftarrow \text{Var}(\{V_{\psi_i}(s_t)\}_{i=1}^m), t = 0, \cdots, T$;
    Store all visited states and uncertainty
      $\mathcal{M} \leftarrow \{(s_t, w_t)\}_{t=0}^T$;
    $\mathcal{Q} \leftarrow$ **Update**$(\mathcal{Q}, \mathcal{M})$;    // Alg. 2
**until** $\pi_\theta$ converges;

**Algorithm 2: State buffer update rule**

**Input:** State archive $\mathcal{Q}$ of capacity $K$; new
 state buffer $\mathcal{M}$.
// Update old density
**for** $(s, w) \in \mathcal{Q}$ **do**
    Recompute $w$ using the latest value
      ensemble $w \leftarrow \text{Var}(\{V_{\psi_i}(s)\}_{i=1}^m)$;
**for** $(s, w) \in \mathcal{M}$ **do**
    **if** $\min_{(s_i, \cdot) \in \mathcal{Q}} d(s_i, s) > \delta$ **then**
      $\mathcal{Q} \leftarrow \mathcal{Q} \cup \{(s, w)\}$;
      **if** $|\mathcal{Q}| \geq K$ **then**
        $j \leftarrow \arg\min_i w_i, (s_i, w_i) \in \mathcal{Q}$;
        $\mathcal{Q} \leftarrow \mathcal{Q} \backslash \{(s_j, w_j)\}$;

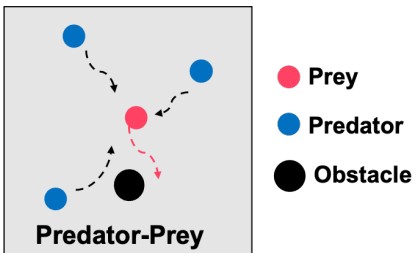

**Figure 2:** Illustration of the *Predator-Prey* environment in MPE.

| Algo. | Best response |
|---|---|
| ZACL | **96.1** (7.6) |
| R-NaD | 109.0 (7.6) |
| FSP | 114.9 (7.7) |
| PFSP | 109.2 (10.1) |
| PSRO | 306.4 (8.1) |
| PSRO (3x) | 176.6 (9.2) |

Table 1: Exploitability of policies learned by different methods. We report the best response to the fixed policies of the prey trained with different algorithms at the same environment steps. Lower number is better.

state generator otherwise. To ensure the state buffer has good coverage of the whole state space, we only insert the new visited states that are at least $\delta$ distance apart from every existing state in the buffer. When the number of states grows larger than the buffer's capacity, we simply delete some states with the lowest value uncertainty. The overall algorithm is summarized in Alg. 1.

## 5 EXPERIMENT

We consider two environments, the multi-agent particle-world environment (MPE) (Lowe et al., 2017) and the hide-and-seek environment (HnS) (Baker et al., 2020). Every experiment is repeated over 3 seeds. More environment and training details can be found in Appendix B.

### 5.1 THE MULTI-AGENT PARTICLE-WORLD ENVIRONMENT

We first test the performance of ZACL in the *Predator-Prey* environment of MPE. As illustrated in Fig. 2, there are three cooperative predators chasing one prey in a 2D environment with several obstacles blocking the way. The game configurations are the positions of all the agents and the obstacles. We choose the following baselines for solving zero-sum games: three popular self-play methods fictitious self-play (FSP) (Heinrich et al., 2015), prioritized fictitious self-play (PFSP) (Vinyals et al., 2019) and R-NaD (Perolat et al., 2022), and a population-based training method policy-space response oracles (PSRO) (Lanctot et al., 2017) with a population size of 3. We fix the policies of the prey trained at 15M environment samples for all the methods, then train the best response policy of the predators to measure the exploitability. We additionally try allowing 45M (3 times) samples for PSRO since PBT typically requires more samples. The average episodic reward of the predators evaluated at convergence and the standard deviation over 3 seeds are shown in Table 1. The exploitability of our agent is the lowest, indicating the most competitive performance of our policy.

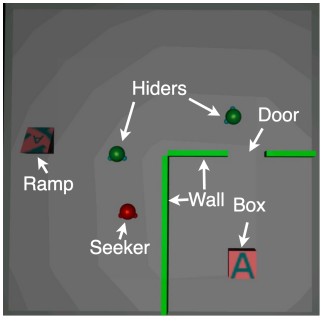

Figure 3: Illustration of a quadrant scenario in hide-and-seek environment.

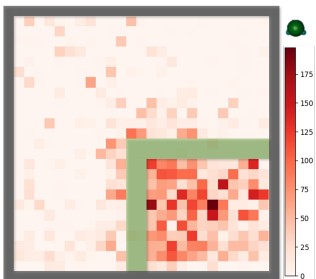
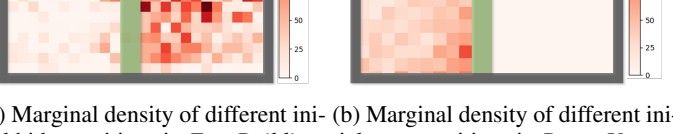

(a) Marginal density of different ini- (b) Marginal density of different initial hider positions in *Fort-Building*. tial ramp positions in *Ramp-Use*.

Figure 4: Density map from which to sample initial game configurations in different training stages. Darker means higher density.

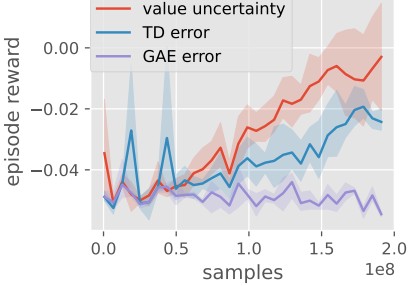

Figure 6: Ablation on the progression metric. Our value uncertainty metric clearly outperforms absolute TD error and GAE error metrics.

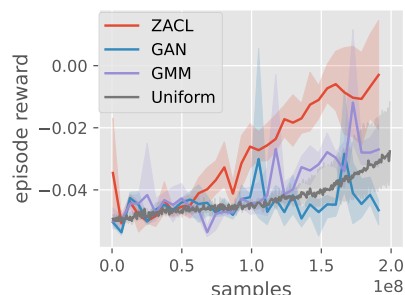

Figure 7: Ablation on different state samplers. Our non-parametric state generator outperforms parametric and uniform baselines.

## 5.2 THE HIDE-AND-SEEK ENVIRONMENT

We consider the fully-observable *quadrant* scenario in the hide-and-seek environment (HnS) as shown in Fig. 3. The scenario consists of a room in the lower right corner which is connected by one door to the remainder of the environment. Two hiders are spawned uniformly in the environment, but one seeker is only spawned outside the quadrant room. One box is spawned inside the room and one ramp is spawned outside the room. Both the box and the ramp can be locked in place. We need to train policy for hiders to avoid lines of sight from the seekers and for the seeker to keep vision of the hiders. The initial states contain the positions of the agents, the box, the ramp, and the door.

As there are a total of 4 rounds of emergent strategies in HnS (Baker et al., 2020), i.e., *Running and Chasing*, *Fort-Building*, *Ramp-Use*, and *Ramp-Defense*, we evaluate the training efficiency of different methods by comparing the number of environment interactions required to master the final round of strategy in Fig. 5. We compare ZACL on a single desktop machine with MAPPO using different batch sizes on a distributed system. Note that the distributed system uses a massive parallel of 9600 environment workers, and divides the collected data into batches of size 64k or 128k for RL optimizing, while ZACL desktop training only uses 256 parallel workers to collect samples until the GPU memory is full in each iteration, then runs RL over all these data as a single batch. ZACL consumes 2.0 billion samples, ~3X more efficient than MAPPO without curriculum. We visualize the value

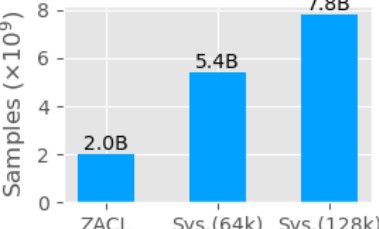

Figure 5: Number of samples required to converge to the strategy of the final stage in HnS. ZACL trained on a single desktop machine is significantly more sample efficient than MAPPO on a distributed system with large batch.

uncertainty of different states in two training stages in Fig. 4 to see how our curriculum samples initial configurations. Fig. 3a depicts the stage when the hiders are about to learn to build a fort. The most uncertain states are where the hider is initialized inside the fort. When the seekers are learning to use the ramp (Fig. 3b), the most uncertain states are the ramp initialized close to the walls.

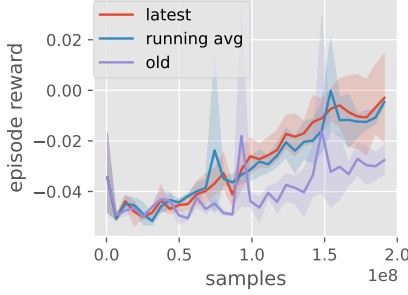

Figure 8: Comparison between assigning the latest, a running average of historical or old value variance to the density in $\mathcal{Q}$.

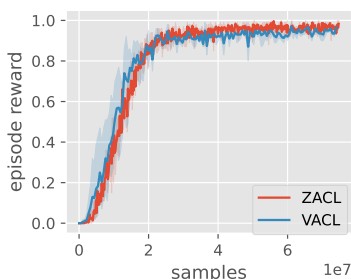

Figure 9: Seeker's average episode rewards in a goal-reaching *Ramp-Use* task. ZACL performs comparable to a strong CL baseline VACL, which is specialized for goal-oriented problems.

## 5.3 ABLATION STUDIES

The experiments in this part are conducted in the HnS environment. All the methods start training from the same checkpoint right after the *Fort-Building* stage. In each plot, we report the seeker's average episode reward w.r.t. the number of environment samples.

**Comparison with other progression metrics:** Our value uncertainty progression metric is compared against two other reward-agnostic metrics: absolute temporal difference (TD) error $|\delta_t| = |R(s_t, a, s_{t+1}) + \gamma V(s_{t+1}) - V(s_t)|$ and absolute GAE (Schulman et al., 2015) value loss $|\sum_{k=t}^{T} (\gamma \lambda_{\text{GAE}})^{k-t} \delta_k|$, and they are implemented with a single value function. The results in Fig. 6 show that the agents with value uncertainty metric learn more efficiently than the baselines.

**Comparison with different state generators:** We substitute our particle-based state sampler with other generators including a neural generator GoalGAN (Dendorfer et al., 2020), a parametric Gaussian mixture model (Portelas et al., 2020) and a naive uniform state sampler. As shown in Fig. 7, parametric generators (GAN and GMM) do not learn as efficiently as our non-parametric method, since they require a large number of samples to get accurate estimation. The uniform state sampler serves as a baseline to show the effectiveness of any curriculum.

**Ablations on the state buffer $\mathcal{Q}$:** To validate the necessity of updating the density in $\mathcal{Q}$ with the latest value variance, we try two variants shown in Fig. 8: updating with the average of stored and new variance (blue), i.e., $w \leftarrow (w + \text{Var}(\boldsymbol{V}(s)))/2$, and keeping the old density (purple). Using running average performs on par to using the latest density, while using staled density significantly hurts the performance. The requirement on keeping track of the latest value uncertainty can also explain the necessity to adopt a fast non-parametric task generator.

## 5.4 DEPLOYMENT IN COOPERATIVE SCENARIOS

Although ZACL is motivated from zero-sum games, it is generally applicable to cooperative tasks. We consider a *Ramp-Use* task proposed in VACL (Chen et al., 2021), where the seeker aims to get into the lower-right quadrant (with no door opening) but is only possible using a ramp. We adopt the same prior knowledge of "easy tasks" used in VACL to initialize $\mathcal{Q}$ and achieve comparable sample efficiency with VACL (see Fig. 9), one of the strongest CL algorithms for cooperative games.

## 6 CONCLUSION

We present ZACL, a theoretically grounded automatic curriculum learning framework for accelerating multi-agent RL training in zero-sum games. We propose to use value uncertainty as a reward-agnostic metric to prioritize sampling different game configurations. With this initial state sampler, the RL agent can practise more in the most uncertain tasks with respect to the learning policy throughout training, thus boosting training efficiency. We report appealing experiment results that ZACL efficiently discovers all emergent strategies using a single desktop machine in the challenging hide-and-seek environment that were originally produced with large-scale distributed training. We hope ZACL can be helpful to speeding up prototype developing in multi-agent RL, and help make RL training on complex problems more affordable to the community.

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

# A    METHOD DETAILS

## A.1    FIRST ORDER APPROXIMATION OF COVARIANCE

Denote $M = \mathbb{E}[Q_\pm]$ as the empirical mean. We can perform second-order Taylor expansion as for the covariance of two random variables $X$ and $Y$ as follows:

$$\text{Cov}[f_1(X), f_2(Y)] \approx f_1'(\mathbb{E}(X))f_2'(\mathbb{E}(Y)) - \frac{1}{4}f_1''(\mathbb{E}(X))f_2''(\mathbb{E}(Y))\text{Var}[X]\text{Var}[Y] \quad (5)$$

Applying $X = Y = Q$, $f_1 = \exp$ and $f_2$ as identity function, we have that

$$\text{Cov}[\exp(Q), Q] \approx \exp(M)\text{Cov}[Q, Q] = \exp(M)\text{Var}[Q, Q] \quad (6)$$

which relates the covariance between $\exp(Q)$ and $Q$ to the variance of $Q$.

## A.2    PROOFS

**Proposition 1.** *For all policies $\pi_\pm^{(j)}$ and $Q_\pm^{(j)}$ such that $\pi_\pm^{(j)} = \exp(Q_\pm^{(j)})$:*

$$\frac{1}{m}\sum_{j=1}^{m}\pi^{(j)}(a_\pm|s)[Q_\pm^\star(s, a_\pm) - \log\pi^{(j)}(a_\pm|s)] \leq \bar{\pi}(a_\pm|s)[Q_\pm^\star(s, a_\pm) - \frac{1}{m}\sum_{j=1}^{m}Q_\pm^{(j)}(s, a_\pm)].$$

*Proof.* First, from the convexity of $f(x) = x\log x$, we can apply Jensen's inequality to $\pi$:

$$\frac{1}{m}\sum_{j=1}^{m}\pi^{(j)}(a_\pm|s)[\log\pi^{(j)}(a_\pm|s)] \geq \bar{\pi}(a_\pm|s)\log\bar{\pi}(a_\pm|s) \quad (7)$$

Therefore:

$$\frac{1}{m}\sum_{j=1}^{m}\pi^{(j)}(a_\pm|s)[Q_\pm^\star(s, a_\pm) - \log\pi^{(j)}(a_\pm|s)] \leq \bar{\pi}(a_\pm|s)[Q^\star(s, a_\pm) - \log\bar{\pi}(a_\pm|s)]. \quad (8)$$

Then, from the concavity of $g(x) = \log x$ we can apply Jensen's inequality to $Q$:

$$\log\bar{\pi}(a_\pm|s) = \log\frac{1}{m}\sum_{j=1}^{m}\exp(Q_\pm^{(j)}(s, a_\pm)) \geq \frac{1}{m}\sum_{j=1}^{m}Q_\pm^{(j)}(s, a_\pm). \quad (9)$$

Thus:

$$\frac{1}{m}\sum_{j=1}^{m}\pi^{(j)}(a_\pm|s)[Q_\pm^\star(s, a_\pm) - \log\pi^{(j)}(a_\pm|s)] \leq \bar{\pi}(a_\pm|s)[Q^\star(s, a_\pm) - \log\bar{\pi}(a_\pm|s)]$$

$$\leq \bar{\pi}(a_\pm|s)[Q_\pm^\star(s, a_\pm) - \frac{1}{m}\sum_{j=1}^{m}Q_\pm^{(j)}(s, a_\pm)]$$

which completes our statement.    $\square$

# B    IMPLEMENTATION DETAILS

## B.1    TRAINING DETAILS

**Predator-Prey**: In this 2D environment, three predators and one prey are randomly generated in a square area with a side length of 1.0. There are also two obstacles uniformly initialized in the scene. The environment is fully observable, with states defined as a concatenation of the positions, velocities and types of all the agents and obstacles. At each timestep, the agents can choose to move one unit from four different directions or take an idle action. Whenever a predator catches the prey, all predators get +10 reward and the prey gets -10 reward. We adopt a simple MLP architecture for PPO training. All hyper-parameters for training are listed in Table 2.

**Quadrant scenario in HnS**: The environment is fully observable and the observation is a concatenation of positions and velocities of all the agents, the box, the ramp, and also the indicators representing whether the box and ramp are locked. In this environment, when the hider is spotted, the seeker gets a reward of +1. Otherwise, the hider gets +1. We adopt the same network architecture as (Baker et al., 2020), where the states for each entity, i.e., the agent itself, the other agents, the box and the ramp, are first normalized and encoded with fully connected linear layers, then fused with a self-attention network. We then average-pool entity embeddings and concatenate this pooled representation to $x_{self}$. Finally the pooled-representation is sent into another dense layer and LSTM before outputting actions. All hyper-parameters of are listed in Table 3.

**Non-competitive *Ramp-Use* scenario in HnS**: In this scenario, we use the same network architecture and prior knowledge for defining easy tasks as (Chen et al., 2021). The initial states in $\mathcal{Q}$ have the ramp right next to the wall, and agents located next to the ramp. All hyper-parameters are listed in Table 4.

Table 2: ZACL hyper-parameters used in the particle-world environment.

| Hyper-parameters | Value |
|---|---|
| Learning rate | 5e-4 |
| Discount rate ($\gamma$) | 0.99 |
| GAE parameter ($\lambda_{\text{GAE}}$) | 0.95 |
| Gradient clipping | 20.0 |
| Adam stepsize | 1e-5 |
| Value loss coefficient | 1 |
| Entropy coefficient | 0.01 |
| Parallel threads | 100 |
| PPO clipping | 0.2 |
| PPO epochs | 4 |
| Horizon | 80 |
| $P_{exp}$ | 0.7 |
| Ensemble size $m$ | 3 |
| State distance threshold $\delta$ | 1.0 |
| Capacity $K$ | 2000 |

Table 3: ZACL hyper-parameters used in the hide-and-seek environment.

| Hyper-parameters | Value |
|---|---|
| Learning rate | 3e-4 |
| Discount rate ($\gamma$) | 0.998 |
| GAE parameter ($\lambda_{\text{GAE}}$) | 0.95 |
| Gradient clipping | 5.0 |
| Adam stepsize | 1e-5 |
| Value loss coefficient | 1 |
| Entropy coefficient | 0.01 |
| PPO clipping | 0.2 |
| Chunk length | 10 |
| PPO epochs | 4 |
| Horizon | 80 |
| Mini-batch size | 57600 chunks of 10 timesteps |
| Size of embedding layer | 128 |
| Size of MLP layer | 256 |
| Size of LSTM layer | 256 |
| Residual attention layer | 32 |
| Weight decay coefficient | $10^{-6}$ |
| $P_{exp}$ | 0.7 |
| Ensemble size $m$ | 3 |
| State distance threshold $\delta$ | 2.0 |
| Capacity $K$ | 10000 |

Table 4: ZACL hyper-parameters of goal-reaching scenarios used in the hide-and-seek environment.

| Hyper-parameters | Value |
|---|---|
| Learning rate | 5e-4 |
| Discount rate ($\gamma$) | 0.99 |
| GAE parameter ($\lambda_{\text{GAE}}$) | 0.95 |
| Gradient clipping | 20.0 |
| Adam stepsize | 1e-5 |
| Value loss coefficient | 1 |
| Entropy coefficient | 0.01 |
| PPO clipping | 0.2 |
| chunk length | 10 |
| PPO epochs | 15 |
| Horizon | 60 |
| Parallel threads | 300 |
| $P_{exp}$ | 0.7 |
| Ensemble size $m$ | 3 |
| State distance threshold $\delta$ | 2.0 |
| Capacity $K$ | 2000 |

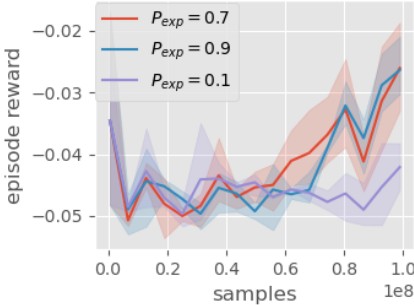

Figure 10: Ablations on the ratio $P_{exp}$ of sampling initial environment states from the diversified buffer. $P_{exp}$ should be sufficiently large to ensure good performance.

### B.2 MORE RESULTS

**Exploration - exploitation trade-off**: The initial states are sampled from a mixture of uniform distribution and estimated value uncertainty density in the main results of hide-and-seek environment. We compare the performance using different ratios $P_{exp} = 0.1, 0.7, 0.9$ to reset from the diversified buffer in Fig. 10. When $P_{exp}$ is too small, the training progress slows down since the agent cannot effectively utilize the curriculum using insufficient high-uncertainty reset states. When $P_{exp}$ is close 1, the performance becomes slightly worse. We set $P_{exp}$ to 0.7 as it empirically achieves good balance between exploration and exploitation.

