# OpenReview forum: "Automatic Curriculum Generation for Reinforcement Learning in Zero-Sum Games"
_ICLR.cc/2023/Conference — Submitted to ICLR 2023_

### Official Review · Reviewer_EQHM · 2022-10-20

**Confidence:** 5
**Correctness:** 3
**Technical Novelty And Significance:** 2
**Empirical Novelty And Significance:** 2
**Recommendation:** 5

**Clarity, Quality, Novelty And Reproducibility:**

Overall the writing is clear. The paper should present the gist of the experiments in the main paper instead of deferring everything to the appendix. The theoretical derivation is correct and the presentation quality is good. The idea of using the value disagreement is not novel, the authors pointed that in the related work. The theoretical formulation for zero-sum games is somehow novel but not significant. No code submission to reproduce the results.


**Strength And Weaknesses:**

The authors haven’t mentioned nor compared to Neural Auto-Curricula [1], which is also a method for automatic curriculum generation in zero-sum games. This is essential given the claim in the abstract that ZACL is the first method to tackle the problem.
In related work, the paper states that the method is faster than emergent behavior methods, this claim is not supported in the results.
In the paragraph after equation 2, the energy-based formulation should be approximate, not exact. Equation 8 the Q function needs +-.
In the experimental section, the comparison to baseline methods was only in the easy MPE environment, while we see only comparisons to MAPPO and ablations in HnS.
It would be useful to report the time of the training time when assuring using a single desktop machine as a major contribution (given the method needs 2.0 billion samples in figure 5).

[1] Feng, X., Slumbers, O., Wan, Z., Liu, B., McAleer, S., Wen, Y., ... & Yang, Y. (2021). Neural auto-curricula in two-player zero-sum games. Advances in Neural Information Processing Systems, 34, 3504-3517.


**Summary Of The Paper:**

The paper presents a theoretical framework for automatic curriculum generation in the setting of zero-sum games using the Q-value variance as an implicit signal of the learning progress toward the convergence criterion (Nash Equilibrium).
The theoretical derivation solves a problem of distribution matching for multi-agent RL, optimizing energy-based policies due to a statistical distance to an LSBRE optimal policy. The proposed method ZACL (Zero-sum Automatic Curriculum Learning) optimizes Jensen's gap between the Q-function of the individual policies and the average policy, which is equivalent to the value disagreement proposed for goal-directed problems in [1].
The variance of the value is approximated using an ensemble of value functions. Tasks are sampled from the value uncertainty distribution using a non-parametric particle-based sampler. The method uses the MAPPO algorithm to train the agents, with some tricks to improve exploration and control the replay buffer.
The experimental section evaluates the method in two environments; MPE and HnS. The results suggest performance gain and verify most of the claims in the paper.

[1] Zhang, Y., Abbeel, P., & Pinto, L. (2020). Automatic curriculum learning through value disagreement. Advances in Neural Information Processing Systems, 33, 7648-7659.

**Summary Of The Review:**

Some of the claims of the paper weren’t evaluated, like the ones mentioned in the comments.
The technical contributions are marginal and not significant, given the previous work on using the value disagreement [1].
The experimental contributions are marginal as well, where the evaluation is similar to the VARL [2] paper.
Given that, the paper is marginally below the acceptance threshold.

[1] Zhang, Y., Abbeel, P., & Pinto, L. (2020). Automatic curriculum learning through value disagreement. Advances in Neural Information Processing Systems, 33, 7648-7659.

[2] Chen, J., Zhang, Y., Xu, Y., Ma, H., Yang, H., Song, J., ... & Wu, Y. (2021). Variational Automatic Curriculum Learning for Sparse-Reward Cooperative Multi-Agent Problems. Advances in Neural Information Processing Systems, 34, 9681-9693.

Post-rebuttal feedback
------------------------------
I have read the revised version and found few significant improvement. Thus, I'll keep my score.

---

> ### Author Response · Authors · 2022-11-14
> **Thanks for your valuable comments. We will put more discussions into the final paper for easier understanding.**
>
> 1. “ We haven’t mentioned nor compared to Neural Auto-Curricula ”
>
> Thanks for your reminder, we will supplement the comparison experiment in the final version. It's important to clarify that we are the first one to evaluate and design the curriculum at the level of subgame, while Self-Play, PSRO or Neural Auto-Curricula design the curriculum at the policy level and train against moderate opponents.
>
> 2. “ The paper states that the method is faster than emergent behavior methods, this claim is not supported in the results. ”
>
> In MPE, we fix the policies of the prey trained at 15M environment samples for all the baselines, then train the best response policy of the predators to measure the exploitability. We additionally try allowing 45M (3 times) samples for PSRO since PBT typically requires more samples. This result indicates that our algorithm has the highest sample efficiency.
>
> 3. “In the paragraph after equation 2, the energy-based formulation should be approximate, not exact. Equation 8 the Q function needs +-. ”
>
> Thank you for pointing out our typos in the theoretical proof.
>
> 4. “ The comparison to baseline methods was only in the easy MPE environment, while we see only comparisons to MAPPO and ablations in HnS ”
>
> We will supplement all experiments in the future. Thank you for your suggestions.

---

### Official Review · Reviewer_5rst · 2022-10-24

**Confidence:** 4
**Correctness:** 3
**Technical Novelty And Significance:** 2
**Empirical Novelty And Significance:** 2
**Recommendation:** 3

**Clarity, Quality, Novelty And Reproducibility:**

I find the work relatively clear (with the issues expanded in the previous section), and of high quality.

I find the novelty of the method itself limited, but I find the proposed task to be of high experimental novelty.

I find the first experiment reproducible, the second one: not so much (again, details in the previous section).

**Strength And Weaknesses:**

## Strengths

The results on hide and seek sound impressive, and the ones for Predator-Prey are evaluated against good baselines.

The proposal of using convergence time on HnS is an interesting idea, and a useful contribution on its own, as it could become a common tool for evaluating curriculum learning methods.

## Weaknesses

I don't find the argumentation behind concentrating on 0-sum (the rewards are not growing) convincing, as there is a lot of work that define progress in 0-sum games this way or another (apart from the cited PSRO-derived ones, eg. [1] claims size of a NE is a useful measure of progress). Furthermore, the mathematical motivation behind the method nor the method itself doesn't use the assumption of the game being 0-sum in any way, which makes the 0-sum part unintuitive.

The mathematical analysis provides a motivation for a significantly different method:

1. One that uses Q functions, and not value functions. There is no explanation provided on why using V provides an accurate approximation to Q. I don't understand why authors couldn't use Q-function (with actions sampled from the policy) instead, following their motivation.
2. A Maximum entropy one.
3. One with a fixed entropy weight, which on its own is a relatively strong assumption in a Curriculum Learning setting, where we may want to decrease entropy penalty over the course of training as the policy gets better and better.

The comparison with MAPPO on HnS doesn't provide enough detail to be properly judged: it's not clear what machine is used in both cases: the phrasing of the paper makes me assume that for ZACL there is a single GPU, able to process 2B samples in a single batch, but that seems beyond the current hardware possibilities. It's not clear why MAPPO uses 9600 workers while ZACL 256 and why the batch sizes are significantly different between the methods. Finally, it would be useful to test one of the other SOTA methods (eg. from the ones tested on Predator-Prey) on this task for a reliable comparison.

As authors correctly cite, a very similar method for optimizing the tasks based on maximizing a Q-value variance was proposed in 2020.

## Future work suggestion

One unaddressed issue of the maximizing-variance method is the variance coming from the environment. If there are tasks which inherently possess such variance (eg. there are two doors chosen randomly and if the agent chooses wrongly, it's not able to get to the goal), they will always be on top of the priority queue, even when with the optimal policy of the agent, wasting environment interaction on tasks where nothing else could be improved.

[1]: Czarnecki et al. [Real World Games Look Like Spinning Tops](https://arxiv.org/abs/2004.09468)

**Summary Of The Paper:**

The paper presents a method for choosing an auto curriculum strategy: a big buffer with all visited states is preserved, and the initial state for starting a trajectory is sampled from it proportionally to the variance of a number of value functions trained jointly with the policy.

The method is motivated by the lack of progression metric in 0-sum games, where the absolute value of the reward doesn't correspond to the progress in training where multiple agents are training in parallel.

Authors also provide a mathematical motivation of using the Q-value variance, under the assumption of using a MaxEnt policy with a fixed $\lambda$ parameter.

The experiments consist of a simple Predator-Prey environment where the proposed method (ZACL) is compared against a number of 0-sum alternatives, and 4 rounds in the hide-and-seek (HnS) environment (where the goal is to converge to a Nash equilibrium as quickly as  possible), where MAPPO is used as a baseline.

**Summary Of The Review:**

Overall, the proposed method is reasonable, but with little novelty. I find the authors' claim on "complete theoretical derivation" of the work to be an overstatement, given that the empirically evaluated method differs significantly from the one being derived. Similarly, I don't find the 0-sum motivation matching the content of the paper.

I find the proposed evaluation task very interesting, but provided results are not detailed enough to be confident that it provides a useful signal to evaluate CL methods on.

edit: after the usage of HnS task was pointed out by another reviewer, I decided to decrease the score from 5 to 3, given the (unchanged) review above, claiming the task is the novel part of the paper.

---

> ### Author Response · Authors · 2022-11-14
> **Thanks for the valuable comments. We promise to add all the clarifications to our final version**
>
> 1. " There is a lot of work that define progress in 0-sum games "
>
>     There is a lot of work to define progress in 0-sum games at the policy level and they concentrate on training against moderate opponents. But we use a completely different metric, which automatically generates a curriculum at the subgame level to accelerate convergence to Nash.
>
> 2. " The mathematical motivation behind the method nor the method itself doesn't use the assumption of the game being 0-sum "
>
>     The advantage of our curriculum method is that it is not limited to 0-sum games. As mentioned in the article, we can solve fully cooperative games and accelerate the training process.
>
> 3. “ The mathematical analysis provides a motivation for a significantly different method ” \
>     ( 1 ) We use Q functions in the proof, but Q functions and V functions are equivalent in the algorithm implementation. In our implementation, we only need to use the relative value of difficulty as a metric for sampling. We can represent  the variance of Q functions by V functions and the current policy : $Var_{j=1\sim m}[Q^j_{\pm}(s_t, a_t)] = Var_{j=1\sim m}[r^t + \gamma E_{ s_{t+1} \sim \rho_{s} }[V^j_{\pm}(s_{t+1})]] = \gamma E_{s_{t+1} \sim \rho_{s}}[Var_{j=1\sim m}[V^j_{\pm}(s_{t+1})]]$ . So we directly use the V functions that are available in the MAPPO algorithm.
>
>     ( 2 ) We will use the max entropy version of the algorithm for comparison later. Thanks for your suggestion.
>
>     ( 3 ) For the convenience of proof, we use λ = 1, which corresponds to a certain degree of entropy regularization, but the argument is still valid for other λ > 0.
>
> 4. “ 2B samples in a single batch seems beyond the current hardware possibilities ”
>
>     We use a single a100 GPU to train ZACL, and consume 2B samples in the whole training process. But as mentioned in the appendix, the sample in a single batch is 57,600 chunks of 10 timesteps, which takes up all the GPU memory.
>
> 5. “ It's not clear why MAPPO uses 9600 workers while ZACL 256 and why the batch sizes are significantly different between the methods. ”
>
>     Due to the limitation of compute,  ZACL only uses 256 workers to collect data in a single machine. However, MAPPO training on HnS requires a very large number of samples and consumes a long training time. So, like OpenAI, we use the distributed system and run a lot of parallel training workers to speed up the training process. The results show that we verify that MAPPO cannot exceed the performance of ZACL even if using a larger batch size in the distributed system.
>
> 6. “ Future work suggestion ” \
>     Thanks for your advice.

---

### Official Review · Reviewer_L9ZZ · 2022-10-25

**Confidence:** 4
**Correctness:** 2
**Technical Novelty And Significance:** 2
**Empirical Novelty And Significance:** 2
**Recommendation:** 3

**Clarity, Quality, Novelty And Reproducibility:**

I thought the paper was well-written. The novelty is fine: to my knowledge I don't know of any papers looking at curriculum in two-player zero-sum games. The algorithm seems reproducible.

**Strength And Weaknesses:**

I liked that this paper was able to replicate and improve on the results from the OpenAI hide and seek paper with much less compute. However, there are a number of issues with this paper.
First, I don’t think the motivation is there. If you are already doing self play, why do you need a curriculum? Usually the idea is that you need to use a curriculum to scale up to hard tasks, but the point of the OpenAI paper was to show that self play is an automatic curriculum.
Self play isn’t guaranteed to converge to Nash in two-player zero-sum games, and MAPPO is a cooperative RL algorithm
I don’t understand why PSRO is exploitable by a best response. By design, upon convergence PSRO is not exploitable by a best response. How many iterations did you train it? Could you show approximate exploitability over time for all algorithms? Also, comparisons to CFR-based methods such as ESCHER would help the evaluation.


**Summary Of The Paper:**

This paper ZACL, a curriculum method for self play, where the policy is incentivized by the variance of the Q function

**Summary Of The Review:**

Overall, I think the method is not well-motivated, and the experiments do not convince me that it indeed (1) finds a NE, and (2) converges faster than baselines.

---

> ### Author Response · Authors · 2022-11-14
> **We appreciate your valuable comments.**
>
>  1. “ Why do we need a curriculum ? ” \
>         As mentioned in OpenAI's paper, hiders and seekers generate multiple rounds of strategies through self-play and our main contribution is to reduce the number of samples that reach each stage by generating an automatic curriculum that varies from easy to difficult. Self-play or PSRO-style algorithms use an automatic curriculum at the level of policy and only train against moderate opponents. However, our method is to automatically generate curriculum at the subgame level to accelerate the convergence to Nash.
>
> 2. “ Why is PSRO exploitable by a best response ? ”  \
>         PSRO can theoretically converge to Nash, but needs sufficient samples. Therefore, we fix the policies of the prey trained at 15M environment samples for all the baselines, then train the best response policy of the predators to measure the exploitability. We additionally try allowing 45M (3 times) samples for PSRO since PBT typically requires more samples. The results show that our method is faster than baselines.

---

> > ### Comment · Reviewer_L9ZZ · 2022-12-06
> > **Response**
> >
> > Thank you for the response, but I remain unconvinced so I will keep my score.

---

### Decision · Program_Chairs · 2023-01-20

**Decision:**

Reject

**Justification For Why Not Higher Score:**

There was a consensus among the reviewers that the work is not yet ready for publication.

**Justification For Why Not Lower Score:**

N/A

**Metareview: Summary, Strengths And Weaknesses:**

The reviewers agreed that the paper investigates an important setting of curriculum learning in two-player zero-sum games, is generally well-written, and has an interesting experimental evaluation. However, the reviewers pointed out several weaknesses in the paper, and there was a consensus that the work is not yet ready for publication. The reviewers have provided detailed and constructive feedback to the authors. We hope the authors can incorporate this feedback when preparing future revisions of the paper.